# Is Hepatocellular Carcinoma in Fatty Liver Different to Non-Fatty Liver?

**DOI:** 10.3390/nu14183875

**Published:** 2022-09-19

**Authors:** Xuan-vinh Kevin Nguyen, Jason Zhang, Ken Lee Chin, Stephen Bloom, Amanda J. Nicoll

**Affiliations:** 1Eastern Health Clinical School, Monash University, Melbourne 3128, Australia; 2Central Clinical School, Monash University, Melbourne 3004, Australia; 3Eastern Health, Department of Gastroenterology, 8 Arnold Street, Box Hill, Melbourne 3128, Australia

**Keywords:** non-alcoholic fatty liver disease, hepatocellular carcinoma, fatty liver disease, metabolic syndrome, cirrhosis, chronic liver disease, non-alcoholic steatohepatitis, surveillance, obesity, hepatoma

## Abstract

Background: Non-alcoholic fatty liver disease (NAFLD) has become the most prevalent liver disease in Australia and is recognised to play a role in the development of hepatocellular carcinoma (HCC). There are no clear guidelines regarding screening for HCC in NAFLD. The aim of this retrospective study was to compare the characteristics and survival rates of NAFLD-HCC to patients with non-NAFLD-HCC to help guide future research in this area. Methods: A total of 152 HCC patients with either NAFLD (*n* = 36) or non-NAFLD (*n* = 116) were retrospectively analysed from the HCC database and medical records. Chi-square and independent t-test were used to compare baseline characteristics and Kaplan–Meier curves and Cox models were used for survival analysis. Results: Patients with NAFLD-HCC were more likely to be diagnosed due to symptoms rather than through screening, and at an older age, compared with non-NAFLD HCC. The median survival rates were lower in NAFLD-HCC (17.2 months) than in those with non-NAFLD-HCC (23.5 months). Conclusion: There is a rise in the number of HCC cases in patients with NAFLD, and this has significant implications for hepatologists as they are presented with more advanced diseases and have poorer outcomes. Future studies on HCC will need to identify this group earlier in order to have an impact on the HCC survival rate.

## 1. Introduction

Hepatocellular carcinoma (HCC) is the seventh most common cancer and the second leading cause of cancer deaths worldwide [1]. It accounts for 90% of primary liver cancers and occurs more commonly in men than women [2]. The incidence of HCC is increasing globally, with an estimated incidence rate of 2.7 per 100,000 in developed countries and 6.6 per 100,000 in developing countries. Most HCC cases arise in the setting of cirrhosis, with the most common underlying aetiologies being alcohol-related liver disease, chronic hepatitis B (HBV) and hepatitis C (HCV) [1]. While viral hepatitis is the antecedent cause in over 75% of HCC cases globally [3], the diagnosis of non-alcoholic fatty liver disease (NAFLD) has been identified as the most common underlying cofactor for the development of HCC in Western countries, along with alcohol-related liver disease [4]. NAFLD is now the most prevalent liver disease in Australia, and the second highest cause of liver related mortality in Australia, second only to HCV [5]. It affects an estimated 25% of the Australian population and is characterized by the accumulation of fat within the hepatocytes [6]. Approximately one-third of these will have steatohepatitis, the inflammatory and more progressive variant of the disease, which may result in liver fibrosis and cirrhosis [6].

While the incidence of HCC in the presence of NAFLD cirrhosis is considered to be lower than that of viral hepatitis (2.6% versus 4.0%) [7], the rise of NAFLD as the leading cause of liver disease in Australia since 2013 has begun to translate into increased rates of NAFLD-associated HCC [8]. Estimates suggest that up to a third of future HCC burden will be metabolic liver disease in origin. The life expectancy of those with NAFLD-related HCC is postulated to be lower, secondary to the absence of routine surveillance, larger tumour size and advanced age at time of diagnosis [9]. Furthermore, data suggests HCC can develop in the absence of cirrhosis in NAFLD [10]. Therefore, the identification of those that qualify for surveillance is difficult. Currently, there are no clinical recommendations as to whom with NAFLD should undergo surveillance in the absence of cirrhosis, with no known easily identifiable risk factors that may help to advise service delivery and management plans. Therefore, in this pilot study, we primarily aimed to determine the differences between our NAFLD and non-NAFLD-HCC cohorts to help better characterise this disease and to advise the design of future studies of the relationship between NAFLD and HCC.

## 2. Methods

### 2.1. Study Design

All cases of HCC diagnosed over the period between January 2010 and December 2016 from the HCC database were reviewed for evidence of the metabolic syndrome and NAFLD. Data were retrospectively collected from three independent locations: electronic HCC database (Filemaker^®^), hospital electronic medical records (EMR, Cerner PowerChart^®^) and clinical patient folders (CPF^®^ InfoMedix Pty Ltd., Melbourne, Australia 2019). Missing data were obtained from medical correspondence from independent health providers, including general practitioners, pathology and radiological services. All patients were aged 18 years or over, with a diagnosis of HCC made according to the American Association of the Study of the Liver (AASLD) clinical, radiological and/or histological criteria [11]. Participants required a minimum of six months follow-up post HCC diagnosis for inclusion, with the exception of those that died or had a liver transplant within six months of diagnosis.

The underlying liver disease for each case of HCC was reviewed and cross-referenced against the recorded primary disease aetiology from the cancer multi-disciplinary meeting records. A primary liver disease diagnosis of NAFLD (including non-alcoholic steatohepatitis, with or without cirrhosis) was diagnosed according to AASLD practice guidance for NAFLD [12,13]: by evidence of hepatic steatosis on imaging and/or histology; in the absence of alternative cause of hepatic fat accumulation such as significant alcohol consumption, long-term use of a steatogenic medication or hereditary disorder. Liver disease due to viral hepatitis, alcohol consumption or other liver diseases were recorded as documented by the treating physician and confirmed by serological, radiological and histological data where available. The study population was divided into two separate groups based on the aetiology of their underlying liver disease, either (1) NAFLD-related HCC; (2) non-NAFLD-related HCC. Patients with an undocumented disease aetiology or NAFLD in addition to another liver disease were excluded from analysis.

Baseline data included patient the following demographics: age at diagnosis, gender, ethnicity (self-reported); features of the metabolic syndrome: body mass index (BMI), hypertension; presence or absence of cirrhosis and degree of decompensation (Child–Pugh stage). The presence of type 2 diabetes and hyperlipidaemia was also recorded. The tumour characteristics included: the date and method of HCC diagnosis (symptomatic/screening/incidental), Barcelona Clinic Liver Cancer (BCLC) staging, size of largest lesion at diagnosis, initial management and overall survival (months). In addition, therapy for HCC (initial, second, third line, etc.) received was noted, including treatment response measured via mRECIST criteria [14], and overall survival and death (liver or non-liver related). Baseline biochemical data included creatinine, albumin, prothrombin time ratio (INR), haemoglobin, platelet count, white cell count, neutrophils, alpha-fetoprotein (AFP), alanine aminotransferase (ALT), aspartate aminotransferase (AST), gamma-glutamyltransferase (GGT) and alkaline phosphatase (ALP) which were also collected closest to the time of diagnosis. The presence of cirrhosis was established by histology, transient elastography, clinical, laboratory and/or imaging criteria [6].

Baseline data on metabolic syndrome was collected as close to the time of diagnosis of HCC as possible, including high BMI, type 2 diabetes, hyperlipidaemia and hypertension. The presence of hypertension was confirmed via clinician notes on the EMR, or the presence of anti-hypertensive agents recorded in medication charts. BMI data were available in approximately 70% of the patients. Alcohol intake was assessed as number of standard drinks per day, with greater than four standard drinks per day or fourteen standard drinks per week considered consistent with alcoholic liver disease [15]. The presence of diabetes or hyperlipidaemia were not analysed, as they were confounders as part of the diagnosis of NAFLD.

Ethics approval was obtained by the Eastern Health Office for Research and Ethics local reference number LR25/2017.

### 2.2. Statistical Analysis

Continuous variables were expressed as a mean with a 95% confidence interval (CI). Categorical variables were summarised as a percentage. Chi-squared (χ^2^) and independent t-test were carried out to compare baseline characteristics. Overall survival time from diagnosis was the primary study endpoint, which was expressed as both a mean and median. Univariate and multivariable Cox logistic regression analysis was used to determine predictors of overall survival. The Kaplan–Meier survival analysis was used to plot overall survival as a function of time, which excluded patients which failed to undergo follow-up. Cox models were also used for survival analysis. *p* values < 0.05 were considered statistically significant.

## 3. Results

### 3.1. Study Population

A total of 166 new cases of HCC diagnosed across the health service were analysed. Among those, one was removed due to insufficient clinical information and 13 patients who had NAFLD in conjunction with an additional liver disease were also excluded, leaving 152 patients with complete data available for final analysis. Overall, 36 cases had NAFLD-related HCC, while 116 had non-NAFLD-related HCC (Figure 1). The baseline characteristics of both groups are shown in Table 1.

The NAFLD patients were older at the time of diagnosis, more likely to be female and to be diagnosed due to symptoms rather than through screening. As expected, features of the metabolic syndrome were more common in this cohort, with a higher percentage of patients with hypertension and an increased mean BMI. Type 2 diabetes mellitus and hyperlipidaemia were significantly higher in the NAFLD cohort, as expected, as they form part of the non-histological diagnosis of NAFLD according to the AASLD guidelines. The NAFLD cohort was mostly Caucasian (89%), with 8% of patients being of Asian ethnicity. In the non-NAFLD cohort, a higher proportion of those with Asian descent (24%) was observed, reflecting mainly patients with HBV-related liver disease. Other aetiologies of liver disease in the non-NAFLD population included predominately alcoholic liver disease (30%), Hepatitis B (23%), Hepatitis C (16%) and a mixture of Hepatitis C and alcoholic liver disease (25%), as shown in Table 2.

A higher proportion of NAFLD-HCC patients were not cirrhotic at baseline; however, most did have cirrhosis (81%) and the prevalence of cirrhosis between NAFLD and non-NAFLD patients (90%) was not statistically different (*p* = 0.12). In those with cirrhosis, the distribution of the Child–Pugh stage was not significantly different between the two groups. NAFLD-HCC was more likely to present due to symptoms (*p* = 0.046) with a trend towards larger tumours (*p* = 0.20). Despite this, both cohorts had similar distributions of BCLC stages, with the majority in both groups presenting with BCLC-C disease; however, there were more advanced diseases (BCLC D) in the NAFLD cohort.

### 3.2. Treatment of HCC

The NAFLD-HCC patients received less HCC treatments in total. For initial treatment received, a higher proportion of patients in the NAFLD received curative intent therapy (47% vs. 22% *p* = 0.004) with the majority undergoing surgical resection. This is compared to a predominance of drug-eluting bead transarterial chemoembolization (DEB-TACE) (47%) in the non-NAFLD patients versus NAFLD-HCC (17% *p* < 0.001). Nearly one-third (31%) of NAFLD-HCC patients in the study cohort received best supportive care only, compared with only 17% in the non-NAFLD-HCC group (*p* < 0.001). Only one patient in the NAFLD-HCC group received a liver transplant, compared to five patients in the non-NAFLD group following initial treatment (Table 3).

### 3.3. Overall Survival Analysis

Overall survival data were available on 146 (96%) of the patients. From January 2000 to December 2016, 84 (58%) patients had died, with deaths equally distributed among the NAFLD group (56%) and the non-NAFLD group (58%). The proportion of liver-related deaths in both groups were also equal to 88%. Patients with NAFLD had a median survival time of 17.2 months, compared with 23.5 months in those with non-NAFLD-related HCC. There was no significant difference in the overall survival between NAFLD and non-NAFLD cohorts at four years on Kaplan–Meier survival analysis (Figure 2). None of the NAFLD cohort were diagnosed as BCLC stage 0, most were stage A or C. Survival was better for BCLC B, BCLC C and BCLC D in the non-NAFLD cohort, perhaps reflecting the older age and greater number of comorbidities. The good survival rate of the BCLC A NAFLD group may reflect the subgroup without cirrhosis that were amenable to potentially curative surgery. These results are shown in Figure 3 and Table 4.

### 3.4. Predictors of Overall Survival in Patients with NAFLD and Non-NAFLD-HCC

On univariate Cox regression analysis, higher BMI, hypertension and increased age at diagnosis were associated with reduced overall survival with hypertension having the strongest association (OR 4.31, 95% CI 1.95 to 9.55, *p* < 0.001). In contrast, male sex was associated with improved overall survival (OR 0.19, 95% CI 0.07 to 0.49, *p* = 0.001). Age at diagnosis and hypertension remained predictive on multivariate analysis of poorer survival, while male sex was consistently associated with improved survival. Type 2 diabetes and hyperlipidaemia were not examined in the univariate and multivariate analyses as they are confounded with the diagnosis of NAFLD.

## 4. Discussion

To our knowledge, this is the first study to compare the clinical characteristics of NAFLD-HCC to those of non-NAFLD-HCC in an Australian population. Although the sample size is small, it demonstrates that NAFLD-HCC is an increasing problem in Australia and comprises 24% of the HCC patients managed over the seven-year study period. It is not surprising that NAFLD-HCC occurs in an older cohort, and that these patients are less likely to be on surveillance programs, and hence present symptomatically with larger tumours. In agreement with other studies, the majority of patients were cirrhotic (81%) and male (67%) [16]. This study provides observational information on the nature of NAFLD-HCC, and provides a basis for future work to identify risk factors for HCC in NAFLD patients.

The metabolic syndrome has increasingly been associated with HCC, but the exact relationship remains unclear. The increased BMI of the NAFLD-HCC group may increase the risk of malignancy by placing the body into a chronic inflammatory state via the release of pro-inflammatory cytokines such as interleukin-6 (IL-6) and tumour necrosis factor alpha (TNF-α) [17]. These inflammatory cytokines play a critical role in hepatic fibrinogenesis and this correlation may be secondary to the increased rates of cirrhosis in NAFLD-HCC compared to non-cirrhotic NAFLD. While the pathogenesis behind NAFLD-HCC without cirrhosis is not clear, HCC has been closely associated with type 2 diabetes, an element of the metabolic syndrome. BMI as a marker of poorer survival may represent a surrogate marker for the presence of diabetes. The pro-inflammatory markers TNF-α, IL-6 and insulin-like growth factor (IGF), associated with visceral adiposity and diabetes, are associated in vivo and in vitro models with HCC development and progression [18]. Limited studies suggest improved diabetic control with metformin may reduce the risk of HCC [19]. However, this effect may be related to metformin’s interactions with adenosine monophosphate-activated protein kinase, an apoptotic pathway, in addition to reduction in IGF associated with better diabetic control. A significantly increased prevalence of hypertension and its strong association with poorer survival may suggest an additional mechanism of HCC propagation or may be indicative of the presence of other risk factors for HCC in this cohort. Prior studies have suggested that components of the angiotensin pathway may contribute to cell proliferation, metastasis, reduced apoptosis and immunosuppression [20]. It is not yet known if newer therapies for diabetes that reduce visceral adiposity, NAFLD-specific treatments or certain anti-hypertensives may have an impact on the development and progression of HCC. Increasing evidence supports the fact that HMG-CoA reductase inhibitors (statins) reduce the incidence of HCC.

NAFLD-HCC was demonstrated to have similar four-year survival to HCC with other underlying liver diseases (Figure 2) which is consistent with other studies [21]. This was surprising as NAFLD-HCC patients were older, had more co-morbid diseases, larger tumour burdens and had an increased proportion with advanced diseases (BCLC stage D). It is possible that these tumours have a less aggressive phenotype, but the paucity of liver histology data in all HCC studies makes it difficult to assess this. The comparable survival despite a potentially less aggressive phenotype may be associated with a reduced capacity for multiple sequential therapies due to their age and co-morbidities. The high prevalence of hypertension may make therapy with multi-kinase inhibitors such as sorafenib, lenvatinib and regorafenib more difficult. There were increased rates of curative therapy in this group which we would expect to improve overall survival if successful. However, additional co-morbidities, such as hypertension and increased BMI, may also impact the outcomes of curative intent therapy, in particular surgical resection, where it has been established that an increased number of co-morbidities can affect post-operative complications and overall mortality. Although there was a predominance of male patients in both groups, on univariate and multivariate analysis, male sex was associated with improved overall survival. Unfortunately, similar studies do not comment on or have not shown male sex to be a significant prognostic marker of overall survival and this may reflect the small numbers in our study [22].

NAFLD-HCC is likely to become an even larger subset of patients with HCC in Australia, especially as HBV and HCV treatments and their uptake improves in the era of direct-acting antiviral therapy. Future management strategies for this increasing group of HCC patients needs to be developed to improve outcomes. NAFLD is often silent and overlooked, even in the presence of early cirrhosis, and raising awareness on the risk of HCC in these patients is needed. However, the focus should remain on those with NAFLD cirrhosis, as our data and the data from the United Kingdom have shown this is the highest risk group for HCC development [6,23]. However, further risk stratification is needed in those without cirrhosis, as this group can represent up to 50% of NAFLD-HCC [3,24]. The presence of hypertension and elevated BMI, as seen in our analysis, may be such surrogate markers of risk to identify individuals for screening. Previous studies support this notion, with hypertension and diabetes representing independent risk factors for HCC in those without cirrhosis [25]. Reports are emerging that show that certain genotypes in NAFLD have a greater risk of HCC [26,27].

Most HCC screening programs utilize liver ultrasound as the main tool, which is not always practical for patients with a higher BMI, and this may need to be addressed in the future [28]. The complimentary screening tool, AFP, suffers from many limitations, and future biochemical markers, such as IL-6 and TNF-α, may lead to new breakthroughs in screening. There are well-established screening programs for HBV and HCV in Australia, but not for NAFLD. Therefore, the diagnosis of NAFLD-HCC occurs at a later stage, with a reduction in curative treatment opportunities. This was not observed within our cohort and may reflect physician bias towards curative treatment in the presence of non-cirrhotic HCC where a resection is preferred despite the older age and increased co-morbidities of this subset of patients [29].

There are several limitations to this study. Firstly, the data were retrospectively collected with unavoidable missing data and inability to conduct patient follow-up. The presence of diabetes mellitus as a baseline characteristic was not reported in this study due to incomplete data, but has been shown in other studies as a relative risk of 2.31 for HCC development [30]. Likewise, the presence of hyperlipidaemia was not regularly recorded, which has previously been negatively correlated with HCC development [25]. Additionally, the sample size of the study reduces the accuracy of analysis between the two patient populations but reveals areas for further analysis before their utility in screening can be validated. However, given the low incidence rate of HCC documented in NAFLD (0.08% per year [31]), this limitation is consistently observed in the literature [32]. Subset analysis of NAFLD-related HCC was not adequately powered to determine specific prognostic factors associated with survival in this cohort. A significant proportion of patients in our NAFLD cohort developed HCC without cirrhosis (20%); however, due to the small dataset, it is difficult to clearly determine any features that may help identify at-risk populations to screen. Lastly, a lack of histological diagnosis for most NAFLD adds some uncertainty in the underlying diagnosis.

## 5. Conclusions

In conclusion, this is the first study in an Australian population to analyse the different characteristics and survival rates between NAFLD-HCC and non-NAFLD-HCC. NAFLD HCC outcomes may be improved by better surveillance and identification of risk factors to focus screening on the highest risk group in this increasing population with HCC. This study provides informative data for future prospective analyses regarding the role of metabolic risk factors in HCC development and prognosis.

## Figures and Tables

**Figure 1 nutrients-14-03875-f001:**
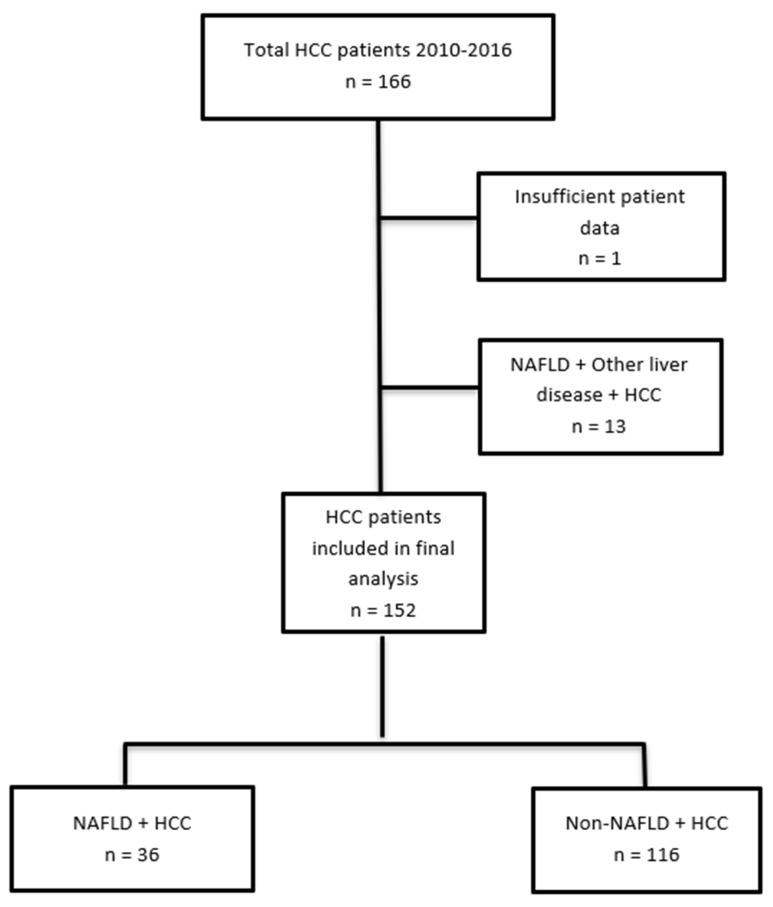
Flow chart of patients included in the analysis and grouping of patients. HCC: Hepatocellular carcinoma; NAFLD: Non-alcoholic fatty liver disease.

**Figure 2 nutrients-14-03875-f002:**
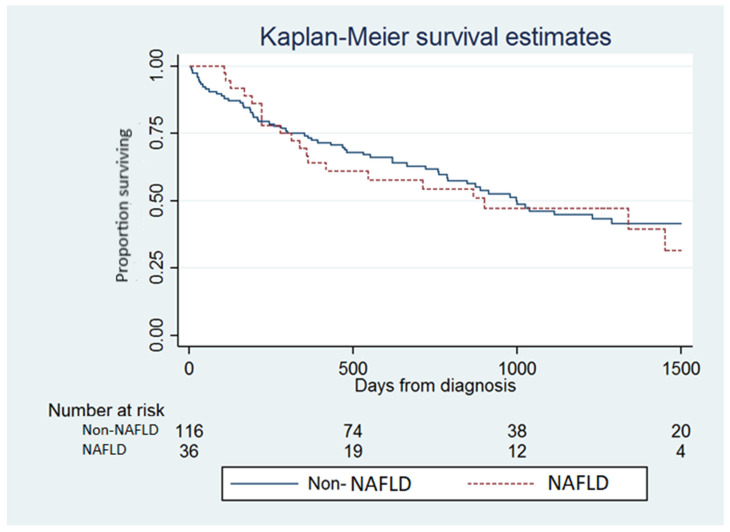
Unadjusted Kaplan–Meier survival curves of patients with NAFLD vs. patients with non-NAFLD. NAFLD: Non-alcoholic fatty liver disease.

**Figure 3 nutrients-14-03875-f003:**
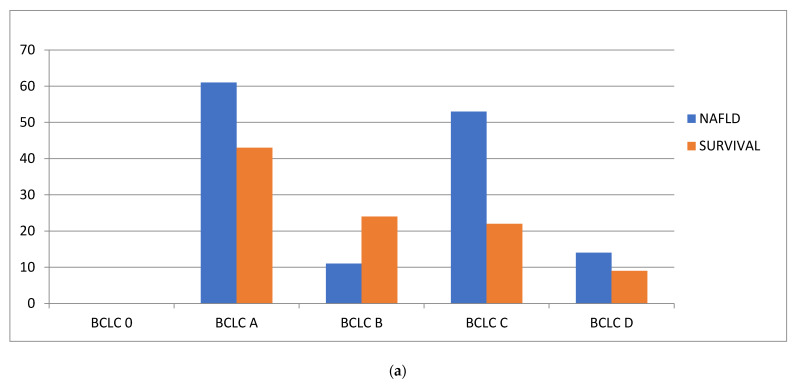
Comparison of the BCLC stage and survival for the (**a**) NAFLD cohort and (**b**) non-NAFLD cohort.

**Table 1 nutrients-14-03875-t001:** Baseline Characteristics of Patients with NAFLD-HCC and non-NAFLD-HCC.

	NAFLD (*n* = 36)	Non-NAFLD (*n* = 116)	*p*-Value
Patient characteristics			
Age at diagnosis, years (mean, 95% CI)	71.9 (69.4–74.3)	62.6 (60.7–64.4)	<0.001
Male, %	24 (66.7)	106 (91.4)	<0.001
Body mass index (mean, 95% CI) (kg/m^2^)	29.1 (26.8–31.4)	26.2 (25.1–27.4)	0.02
Hypertension, %	23 (63.9)	32 (29.1)	<0.001
Type 2 Diabetes	28 (78%)	15 (13%)	<0.01
Hyperlipidaemia	11 (31%)	5 (4%)	<0.01
Ethnicity, %			0.199
Asian	3 (8.3)	28 (24.1)	0.199
Caucasian	32 (88.9)	84 (72.4)	0.199
Hispanic	Nil	1 (0.9)	0.199
Middle Eastern	1 (2.8)	3 (2.6)	0.199
Diagnosis method, %			0.11
Incidental	3 (8.6)	10 (8.9)	0.99
Screening	15(42.8)	69 (61.6)	0.08
Symptomatic	17 (48.6)	33 (29.5)	0.046
BCLC stage at diagnosis, %			0.49
0	Nil	6 (5.2)	0.34
A	8 (22.2)	19 (16.4)	0.46
B	4 (11.1)	17 (14.6)	0.78
C	19 (52.8)	64 (55.2)	0.85
D	5 (13.9)	10 (8.6)	0.05
Cirrhosis, %	29 (80.6)	103 (90.4)	0.12
Stage of cirrhosis, %			0.16
Non-cirrhotic	7 (19.5)	11 (9.6)	0.16
CPA	17 (47.2)	62 (54.4)	0.16
CPB	12 (33.3)	33 (29.0)	0.16
CPC	Nil	8 (7.0)	0.16
HCC characteristics at diagnosis			
Lesion size (mean, 95% CI)	5.4 (3.9 to 6.8)	4.4 (3.7 to 5.1)	0.20
Number of lesions (mean, 95% CI)	1.5 (1.2 to 1.9)	1.7 (1.4 to 1.9)	0.53
AFP (mean, 95% CI)	2882.5 (−1859.0 to 7623.9)	5078.6 (−1204.1 to 11361.3)	0.72

NAFLD: Non-alcoholic fatty liver disease; HCC: Hepatocellular carcinoma; CI: Confidence Interval; BCLC: Barcelona Clinic Liver Cancer; CP: Child–Pugh; AFP: Alpha-fetoprotein. *n t*-test used to compare continuous variables and chi-squared test (χ^2^) used to compare categorical variables between two groups.

**Table 2 nutrients-14-03875-t002:** Underlying liver disease in non-NAFLD-HCC cohort.

Liver Disease	Patient Number (%)
Alcoholic liver disease	35 (30%)
Hepatitis B	27 (23%)
Hepatitis C	18 (16%)
Hepatitis B + Hepatitis C	1 (1%)
Hepatitis B + alcoholic liver disease	2 (2%)
Hepatitis C + alcoholic liver disease	29 (25%)
Hepatitis B + Hepatitis C + alcoholic liver disease	1 (1%)
Autoimmune hepatitis	1 (1%)
Haemochromatosis + alcoholic liver disease	1 (1%)
Primary biliary cholangitis	1 (1%)

NAFLD: Non-alcoholic fatty liver disease; HCC: Hepatocellular carcinoma.

**Table 3 nutrients-14-03875-t003:** Initial HCC treatment in NAFLD and non-NAFLD-HCC patients.

	NAFLD	Non-NAFLD	*p*-Value
Initial Treatment, %			<0.001
Resection	12 (33.3)	11 (9.5)	
DEB-TACE	6 (16.7)	55 (47.4)	
SIRTEX	Nil	6 (5.2)	
RFA	2 (5.6)	12 (10.3)	
MWA	1 (2.8)	2 (1.7)	
Sorafenib	1 (2.8)	9 (7.8)	
Best supportive care	11 (30.6)	20 (17.2)	
Transplant	1 (2.8)	Nil	
DEB-TACE + Resection	Nil	1 (0.9)	
DEB-TACE + RFA	1 (2.8)	Nil	
DEB-TACE + SIRTEX	1 (2.8)	Nil	
Total number of treatments (mean, 95% CI)	1.57 (1.02–2.13)	1.84 (1.55–2.14)	0.192

HCC: Hepatocellular carcinoma; NAFLD: Non-alcoholic fatty liver disease; DEB-TACE: drug-eluting bead transarterial chemoembolization; SIRTEX: SIR-Spheres Yttrium-90 resin microspheres; RFA: radiofrequency ablation; MWA: microwave ablation; CI: Confidence Interval. *t*-test used to compare continuous variables and chi-squared (χ^2^) test used to compare categorical variables between two groups.

**Table 4 nutrients-14-03875-t004:** Survival characteristics of patients with NAFLD-HCC and non-NAFLD-HCC.

	NAFLD	Non-NAFLD	*p*-Value
Deceased, %	20 (55.6)	64 (58.2)	0.78
Cause of death, %			
Liver	15 (88.2)	44 (88.0)	0.98
Non-liver	2 (11.8)	6 (12.0)	0.98
Liver transplant, %	1 (2.8)	5 (4.4)	0.67
Survival from diagnosis, months			
Mean	25.3	30.1	0.23
Median	17.2	23.5	0.47

NAFLD: Non-alcoholic fatty liver disease; HCC: Hepatocellular carcinoma. Chi-squared (χ^2^) test used to compare categorical variables, *t*-test used to compare continuous variables between two groups. Mood’s median test used to compare medians.

## Data Availability

Data may be obtained from the authors.

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
