# Peer review of "Is Hepatocellular Carcinoma in Fatty Liver Different to Non-Fatty Liver?"

_nutrients, 2022, doi:10.3390/nu14183875_

Round 1

Reviewer 1 Report

Despite the authors' efforts to present a work with a high quality, it has major limitations; as said, the authors did not consider some major risk factors for HCC, like diabetes or hyperlipidemia, potentially confounding the results.

Also, how could you definitely confirm the NAFLD cases?

Finally, the sample size was low. So, the accuracy of the findings is questionable.

Author Response

  1. Research Design Must Be Improved

We agree that this is a small retrospective study and therefore is limited like all retrospective studies.  The purpose of the study was to produce data on which to design the prospective study which we are currently doing, and have recruited 330 patients to so far.  This study showed us that data on lifestyle, BMI, alcohol intake and many other features was very difficult to obtain retrospectively and needed to be collected at the initial appointment with the patient.  We are unable to go back and improve the research design but will make the limitations due to the design clearer in the manuscript.

  1. Method Description Must be improved

We have re-written the methods section to improve clarity. The changes are seen in the tracked changes document.

  1. The authors did not consider some major risk factors, like diabetes or hyperlipidaemia

We agree that diabetes and hyperlipidaemia are extremely important considerations in non-alcoholic fatty liver disease. However, as these are features of the metabolic syndrome and were used to identify the cases of NAFLD, it is difficult to assess them in this cohort as this causes their impact to be confounded.  We have made this clearer in the manuscript.  We have added these numbers to Table 1, and of course they are significant, but that is because type 2 diabetes and hyperlipidaemia are part of the diagnosis of NAFLD

  1. How could you definitely confirm the NAFLD cases?

Thank you for this question it is always an important one.  We used the definition in Chalasani et al Hepatology 2018, AASLD clinical guideline for clinical NAFLD. This reference has been added in the methods.  Histological diagnosis was unavailable in most cases in this retrospective study of patients presenting with HCC, but of course histology is still considered the gold standard. The presence of steatosis on ultrasound, consistent biochemistry and features of the metabolic syndrome, with exclusion of other causes of liver injury, are accepted for the diagnosis of NAFLD, as described in Alberti et al Circulation 2009.

  1. The sample size is low. So, the accuracy of the findings is questionable.

We acknowledge that our sample size is low and that this makes it impossible to make strong inferences from the findings. This was a pilot study to inform future prospective work and assisted us in the power calculation for the prospective study, and informed us on the areas that were likely to be problematic in this area.  We have improved the discussion to reflect this.

Reviewer 2 Report

This is a quite important report concerning NAFLD-related HCCs, which tend to be diagnosed at more advanced stages. Many hepatologists have the same idea. This paper has clarified the characteristics of NAFLD-related HCC with living data. Adding a new graph showing the relationship between advanced BCLC stages and worse prognoses in NAFLD-related HCC may help readers' understanding.

Author Response

Response to Reviewer 2:

  1. Presentation of the results can be improved

Thank you for your review of this manuscript and your suggestion, please see below.

  1. This paper has clarified the characteristics of NAFLD-related HCC with living data. Adding a new graph showing the relationship between advanced BCLC stages and worse prognoses in NAFLD-HCC may help the readers’ understanding.

Thank you for your helpful suggestion. A new graph showing the relationship between advanced BCLC stages and poorer prognosis in NAFLD-HCC has been added to the manuscript as Figure 3.

Round 2

Reviewer 1 Report

Thank you for your revisions. Before publication, please reassess the numbers and percentages of tables; e.g., table 1, diagnostic method, NAFLD column: 8.6%+42.9%+48.6%=100.1%! Also, please provide the ethics approval code.

Author Response

Thank you for your advice. We have added the local ethics reference number ot the methods. We have corrected the percentages in Table 1 so that they now add up to 100% not 100.1%. thanks again